# Biodegradable Block Poly(ester amine)s with Pendant Hydroxyl Groups for Biomedical Applications

**DOI:** 10.3390/polym15061473

**Published:** 2023-03-16

**Authors:** Natalia Śmigiel-Gac, Anna Smola-Dmochowska, Henryk Janeczek, Piotr Dobrzyński

**Affiliations:** 1Centre of Polymer and Carbon Materials, Polish Academy of Sciences, 41-819 Zabrze, Poland; 2Faculty of Science and Technology, Jan Długosz University in Czestochowa, 13/15 Armii Krajowej Av., 42-200 Czestochowa, Poland

**Keywords:** biodegradable polymer, aliphatic polyester, functional polymer, wettability, antibacterial polymer

## Abstract

The article presents the results of the synthesis and characteristics of the amphiphilic block terpolymers, built of a hydrophilic polyesteramine block, and hydrophobic blocks made of lactidyl and glycolidyl units. These terpolymers were obtained during the copolymerization of L-lactide with glycolide carried out in the presence of previously produced macroinitiators with protected amine and hydroxyl groups. The terpolymers were prepared to produce a biodegradable and biocompatible material containing active hydroxyl and/or amino groups, with strong antibacterial properties and high surface wettability by water. The control of the reaction course, the process of deprotection of functional groups, and the properties of the obtained terpolymers were made based on ^1^H NMR, FTIR, GPC, and DSC tests. Terpolymers differed in the content of amino and hydroxyl groups. The values of average molecular mass oscillated from about 5000 g/mol to less than 15,000 g/mol. Depending on the length of the hydrophilic block and its composition, the value of the contact angle ranged from 50° to 20°. The terpolymers containing amino groups, capable of forming strong intra- and intermolecular bonds, show a high degree of crystallinity. The endotherm responsible for the melting of L-lactidyl semicrystalline regions appeared in the range from about 90 °C to close to 170 °C, with a heat of fusion from about 15 J/mol to over 60 J/mol.

## 1. Introduction

For many years, bioresorbable polymers have been a valuable material used in medicine and pharmacy in the formation of various types of drug carriers qualifying for their controlled release or in the construction of porous scaffolds used in tissue engineering techniques [1,2,3]. Various kinds of bioresorbable temporary implants (e.g., bonding elements in craniofacial surgery, vascular stents, implants used in orthopedics) or biodegradable surgical threads and non-woven fabrics are also made of this type of polymer [4,5,6]. These most often used bioresorbable biomaterials are primarily aliphatic copolyesters (poly(lactide-*co*-glycolide), poly(lactide-*co*-ε-caprolactone), or poly(ester-*co*-carbonates). There is also an increasing use of this type of polymer in cosmetics, mainly as encapsulation materials for transport care products for the skin [7]. In all these implementations, it is essential to maintain the required high biological purity of the implants or polymer matrices employed. In all these undertakings, it is essential to preserve the required high biological purity of the implants or polymer matrices. This is achieved generally by sterilization with gamma or E-beam irradiation, or plasma chemical sterilization [8]. Unfortunately, there is a high probability of re-infection of these materials during the implantation procedure or use, which should be strongly limited [9]. When it comes to cosmetic products, these products must be biologically pure, but often the secondary effect of cosmetics (dermatologic creams) should be a potent antibacterial activity or the possibility of a significant decrease in the number of these microorganisms. Typical protection against the invasion of bacteria during the treatments is primarily the use of antibiotics, both by administering this drug to patients and by introducing an antibacterial substance into the implanted device itself or on its surface [10]. In cosmetics, some preservatives have been used as antibacterial substances introduced into the composition of manufactured products and formulations, most often parabens [11]. Both the widespread use of antibiotics and parabens or other antibacterial substances such as triclosan (TCS) and triclocarban (TCC) carry many risks associated with the formation of pathogenic bacteria with high drug resistance [12] and with the adverse effect of the listed antibacterial substances on the patient’s health [13]. A much more rational solution seems to be usage of biodegradable polymers, which simultaneously have a strong antibacterial effect.

A typical polymer exhibiting antibacterial properties contains cationic groups that allow its adsorption on the anionic membrane and affect the deterioration of the cell membrane by interfering with the transport of compounds and hydrophobic groups that insert into and disrupt the bilipid membrane [14,15]. Polylactide itself or its copolymers indeed have a noticeable bacteriostatic effect [16]. However, it is insignificant and limited. It can be strengthened by the addition of nanoparticles of selected metals [17] or compounds with strong antibacterial properties [18]. Such a similar effect can also be obtained already at the stage of synthesis of aliphatic polyesters, through the selection of an appropriate bactericidal initiator of polymerization [19].

Several biodegradable natural polymers belonging to the group of polysaccharides, such as chitosan [20] or its derivatives [21,22] and carrageenan [23] preserve a relatively high antibacterial effect. Numerous polypeptides with a pronounced antibacterial impact have revealed antioxidant, self-renewal, and pro-collagen effects, which are desirable in anti-aging cosmetics where equivalent effects are demonstrated [24].

Biomimetic synthetic polymers imitating the chain structure of the host defense peptides (HDPs) have similar properties [14,25]. These polymers contain side amino cations groups. For biocompatibility and biodegradation properties of this type of polymer, the main chain should consist of aliphatic units linked by ester or amide bonds [26]. Polyamines are another group of polymers with a potential bactericidal effect, whose main chain is built of units containing secondary amine groups [27,28]. These polymers were obtained via enzyme-catalyzed copolymerization of the lactone with diethanolamine with either an alkyl or an aryl (phenyl) substituent on the nitrogen. Due to the lipase activity in this reaction, the monomer could be ε-caprolactone, or large ring lactones such as 12-dodecanolide, 15-pentadecanolide, or 16-hexadecanolide [29]. In this way, polyesteramines containing tertiary amines were obtained [30]. Due to the presence of branches of the tertiary amino groups, it was mostly a product in the form of a viscous liquid. The obtained terpolymers were used in research on the possibility of employing them in gene delivery methods. Unfortunately, the publication’s authors did not study these compounds’ bactericidal activity [31]. Similar linear polymers contained in the chain secondary amine groups, with improved anti-staphylococcal activity, in a two-step reaction were manufactured [28]. In the first stage, 1,12-diaminododecane was synthesized by reaction with two equivalents of acrylonitrile to obtain the dinitrile. In the second step, the nitrile functional groups were reduced to primary amines using catalytic Raney nickel. The polymers showed a strong antibacterial effect, also against *S. aureus* strains resistant to natural polyamines. This polymer has also been shown to have a strong synergistic effect with daptomycin, oxacillin, and vancomycin. However, due to the lack of ester bonds and the low content of amide bonds, the described polyamine is very slowly biodegradable.

In the research described in this publication aimed at obtaining biodegradable and biocompatible polymers with strong antibacterial and antifungal properties, we tried to acquire linear copolymers containing secondary amines in the chain. For research, we selected polyesters due to their well-tested and known biodegradation and biological properties. In addition, to extend its usefulness in biomedical applications, we prepared amphiphilic polymers with a block structure showing improved wettability and relatively fast bioresorption. A material with such properties is attractive for many tissue engineering applications or as an antibacterial component of cosmetic or dermatological formulations containing biologically active substances, being able to act as a micro- or nanocarrier of selected drugs. When planning the method of obtaining polyesters with the above-mentioned properties, we were particularly interested in the block ones obtained in ROP of L-lactide initiated with macroinitiators found in the polycondensation of tartaric acid derivatives with previously protected hydroxyl groups and aliphatic diols [32,33]. Oligomers obtained in the reaction of polytransesterification of tartaric acid ester containing protected hydroxyl groups were also used in the formation of polyurethane with polyester blocks [34]. Unfortunately, all of these polyesters did not contain pendant hydroxyl groups. Muñoz-Guerra et al. used the method described previously, using a macro-initiator obtained from a derivative of tartaric acid and butanediol, synthesized a triblock polyester, which contained pendant hydroxyl groups, after deprotection with trifluoroacetic acid (TFA). Unfortunately, the masses of the obtained copolymers were low, amounting to only a few thousand g/mol. These copolymers did not have antibacterial properties; on the contrary, they were a suitable medium for them [35]. After the protection of methoxy groups, the tartaric acid derivative was used in the reaction with diamines to obtain a series of polyesteramides. In this case, the authors did not obtain a polymer with pendant OH groups too [36].

Using the methods described overhead, we used tartaric acid to obtain a new group of biodegradable block co-polyesters containing side hydroxyl and secondary amine groups, potentially meeting the previously set requirements. The publication presents issues related to the synthesis and research of the physicochemical properties of the obtained block polyesteramines. Antibacterial properties, results of cytocompatibility tests, as well as the course of biodegradation of the obtained terpolymers, will be presented later in a separate publication.

## 2. Materials and Methods

### 2.1. Materials

Monomers: l-Lactide, Glycolide (Huizhou Foryou Medical Devices Co., Ltd., Guangdong, China), 1,4-butanediol BD (Aldrich, Steinheim, Germany), Diethanolamine DEA, *N*,*N*′-Bis(2-hydroxyethyl) ethylenediamine HEDA, (+)−Dimethyl L-tartrate (Merck KGaA, Darmstadt, Germany) were used as received.

Initiators and catalysts: zirconium (IV) acetylacetonate, zinc (II) acetylacetonate monohydrate (Aldrich, Steinheim, Germany), and titanium (IV) butoxide (Alfa Aesar, Haverhill, MA, USA) were used as received.

Auxiliary substances: 2,2-Dimethoxypropane, Di-*tert*-butyl dicarbonate, Toluene-4-sulfonic acid monohydride (Merck KGaA, Darmstadt, Germany), tetrahydrofuran, anhydrous (Avantor, Gliwice, Poland).

### 2.2. Synthesis of Dimethyl 2,3-O-Isopropylidene-l-tartrate

The hydroxyl groups of the tartaric acid ester were protected by a reaction with dimethoxypropane (Figure 1). The method used was a modification described earlier [37].

In a round-bottom flask with a capacity of 500 mL equipped with a magnetic stirrer was dissolved, in anhydrous chloroform (50 mL), tartaric acid methyl ester (*n* = 0.336 mol) and then the blocking agent 2,2-dimethoxypropane (1.094 mol) and p-toluenesulfonic acid (*n* = 0.0336 mol) (3%).

The highest yield, over 70%, was obtained using an excess of dimethoxypropane in the weight ratio to tartaric acid ester up to 3:1, carrying out the reactions for 24 h. The reaction course was monitored on an ongoing basis with the help of FTIR and periodically with the use of ^1^H NMR spectrometry (Appendix A). The signal appearing at δ = 4.9 ppm A’ indicated the blocking of the OH groups. Based on the intensity of the protons −CH-OH (signal A δ = 4.5 ppm) and the emerging signal A’ (δ = 4.8–4.9 ppm), efficiency of the reaction was estimated. The final product was diluted with ethyl acetate and washed several times with cold aqueous KHCO_3_ to remove unreacted materials and catalysts. A tartaric acid ester derivative with blocked hydroxyl groups is very sparingly soluble in water. After the separation of the organic phase, the solvent was evaporated from it. The purified compound was dried with MgSO_4_.

The product’s purity level was checked by TLC thin-layer chromatography on a silica gel plate. The positions of the compounds were determined by placing the plate under the light of a UV lamp with a wavelength of 254 nm.

### 2.3. Synthesis of Tert-Butyl Bis(2-hydroxyethyl) Carbamate (Boc-DEA) and N,N′-Ethane-1,2-diylbis[N-(2-hydroxyethyl)-2,2-dimethylpropanamide](Boc_2_-HEDA): Procedures for Blocking Secondary Amine Groups in Diethanolamine and N,N′-bis (2 Hydroxyethyl) Ethylenediamine

Blocking of secondary amine groups of diethanolamine (DEA) and *N*,*N*′-bis (2 hydroxyethyl) ethylenediamine (HEEDA) was carried out by reaction with di-tert-butyl di-carbonate (Boc)_2_O (Figure 2) according to a modified procedure described earlier [38].

In a two-neck flask with a capacity of 250 mL equipped with a reflux condenser and magnetic stirrer, 10.51 g of (0.01 mol) diethanolamine (DEA) was dissolved in 40 mL THF. After dissolving DEA, (Boc)_2_O 22.83 g (0.01 mol) solution in 10 mL THF was added dropwise with vigorous stirring. After adding all of the elements, reactions were continued for 5 h at 40 °C. Then, THF and formed t-butanol were distilled off in a vacuum evaporator. A viscous liquid diethanolamine with a blocked secondary amine group was created with an efficiency of about 90%. During the blocking of secondary amino groups, characteristic b’ signal (δ = 3.39 ppm) (CH_2_)_2_N(CO)O- protons were observed with the simultaneous disappearance of the primary b signal of the –(CH_2_)_2_NH- group protons around δ = 2.75 ppm (Appendix A). Another slight signal shift is observed from δ = 3.69 ppm of the signal a −CH_2_CH_2_OH to δ = 3.77 pmm, and a new appearance of the methyl group signal Boc at δ = 1.46 pmm. The phenomenon proves the block of amino groups and the formation of carboamines.

Blocking of secondary amino groups in *N*,*N*′-bis (2 hydroxyethyl) ethylenediamine HEDA was performed in the same way as in DEA. Reactions were carried out in a stoichiometric molar ratio of HEDA:DDC such as 1:2. The yield of the reaction was also about 90%.

When blocking secondary amino groups *N*,*N*′-bis(2 hydroxyethyl) ethylenediamine, a shift of the E signal from the NH-CH_2_-CH_2_-NH- groups (2.04 ppm) and from the protons of the CH_2_CH_2_NH- groups at 2.79–2.78 ppm (F) to signals in the range of 3.47–3.38 pmm (E’, F’) were observed. The G signal of HO-CH_2_- groups at 3.66 ppm shifted to 3.73 ppm (G’). The g signal in the range of 1.48–1.45 ppm comes from methyl groups.

### 2.4. Synthesis of Polyester Macroinitiators; Poly(butylene 2,3-O-isopropylidene-l-tartrate), Poly(tert-butyl-bis(2-hydroxyethyl) Carbamate 2,3-O-Isopropylidene-l-tartrate) and Poly(N,N′-ethane -1,2-diylbis[N-(2-hydroxyethyl)-2,2-dimethylpropanamide,3-O-Isopropylidene-l-tartrate)

The macroinitiators were obtained by the modified two-step polytransesterification reaction conducted in bulk. The synthesis was carried out in a round-bottom flask with a volume of 0.25 dm^3^, heated and equipped with a distillation system, a thermometer, and an efficient mechanical stirrer. The vessel had connections to argon and the vacuum pump. The first stage of the synthesis consisted of transesterification of a mixture of dimethyl 2,3-O-Isopropylidene-l-tartrate (DIPT) with 1,4-butanediol (BD), or tert-butyl bis(2-hydroxyethyl) carbamate (Boc-DEA), or *N*,*N*′-ethane-1,2-diylbis[N-(2-hydroxyethyl)-2,2- dimethylpropanamide] (Boc_2_-HEDA). In the reaction mixture started, the molar ratio of diols to acid ester was 1.07:1. Constantly stirring the reactor contents at 130 °C, the Ti(OC_4_H_9_)_4_ catalyst was added in an amount of 0.4 wt.%. Arising during the reaction, methyl alcohol was removed by distillation. In the second stage of the synthesis, after gradually reducing the pressure (to about 50 mbar), the temperature of the reaction mixture was progressively increased—at a rate of about 10 °C every 30 min until 150–160 °C. At this temperature, the reaction was continued, controlling changes in viscosity and analyzing ^1^H NMR spectra of the sampled reaction mixture. The process was carried out until the assumed molar mass—about 5000–10,000 g/mol of the product was obtained, which was dependent on the reaction time (Figure 1). After approximately 10–12 h, the reaction was finished.

All obtained products were dissolved in chloroform, precipitated in diethyl ether, and then dried in a vacuum at 40 °C to a constant weight.

### 2.5. Copolymerization of L-Lactide with Glycolide in the Presence of Macroinitiators

The terpolymers were obtained by the ring-opening copolymerization reaction of L-lactide and glycolide cyclic monomers in the presence of previously obtained oligomers, terminated with hydroxyl groups and with protected amine and side hydroxyl groups. The same molar ratio of l-lactide to glycolide as 85:15 was used in all syntheses. As a catalyst zirconium (IV), acetylacetonate was used with a molar ratio initiator to monomers as 1:800. The resulting terpolymers were purified by dissolution in chloroform and precipitated in cold methanol. The product was dried to constant weight in a vacuum. The synthesis process was shown in the following example for the preparation of the terpolymer.

Into a dry 0.25 dm^3^ round-bottomed flask fitted with a connection to the argon source and a vacuum pump, with a mechanical stirrer, and heated on an oil bath, 0.1 mol of l-lactide (14.4 g) and 0.018 mol (2.1 g) of glycolide were added under the protection of the argon cushion. After both monomers melted at 110 °C, stirring was started and then poly(butylene 2,3-O-Isopropylidene-l-tartrate)-(oligomer I1) (Table 1) was added in portions. The reaction was carried out for about 3 days.

### 2.6. Conditions and Course of Deprotection of Amino and Hydroxyl Groups in Copolymers

The conditions for simultaneous deprotection of hydroxyl and secondary amino groups in the obtained terpolymers by simple acidic hydrolysis of the carbamide and isopropylidene groups were optimized, based on literature data [33,39]. This reaction was carried out in a THF solution at room temperature by reaction with trichloroacetic acid (TFA) in the presence of water. After about 20 min, the product was precipitated in diethyl ether and dried. The degree of deprotection was determined by observing changes in the ^1^H NMR spectrum, before and after the deprotection of hydroxyl and amino groups. In addition, the presence and amount of hydroxyl groups were determined based on the analysis of ^1^H NMR spectra after reacting the product with trichloro isocyanate.

### 2.7. Measurements

#### 2.7.1. Nuclear Magnetic Resonance (NMR) Spectroscopy

The composition of the polymers was determined with NMR measurements. Based on the observation of the number of end groups on the ^1^H NMR spectrum, the number average molecular weight of these compounds was also determined. The ^1^H NMR spectra of the copolymers were recorded at 600 MHz with the Advance II Bruker Ultrashield Plus Spectrometer (Billerica, MA, USA) and with the use of a 5 mm sample tube. Deuterated DMSO was the solvent, and tetramethylsilane was used as the internal standard. All ^1^H NMR spectra were obtained with 32 scans, a 2.65 s acquisition time, and an 11 ms pulse at 26 °C. The 13C NMR spectra of the synthesized polymers were recorded at 150 MHz, using the same spectrometer and conditions as the proton spectra. The acquisition time was 0.9 s, the pulse width was 9.4 ms, the delay between pulses was 2 s, and the spectral width was 36,000 Hz.

#### 2.7.2. Thermal Properties

By differential scanning calorimetry (DSC, DuPont 1090B apparatus calibrated with gallium and indium) thermal properties, such as glass transition temperatures and heats of melting and crystallization of obtained copolymers, were examined. The glass transition temperature was determined with a heating and cooling rate of 20 °C/min in the range between −100 and 220 °C, according to the ASTM E 1356-08 standard.

#### 2.7.3. Fourier Transform Infrared (FTIR) Spectroscopy

The spectra were recorded in KBr discs in the range of 4000–400 cm^−1^ at 64 scans of samples using a JASCO FT/IR-6700 spectrophotometer (Easton, MD, USA) with a resolution of 2 cm^−1^.

#### 2.7.4. Measurement of Average Molecular Mass and Mass Dispersion

The average number of molecular mass (Mn) and the mass dispersion coefficient (Đ) were determined by gel permeation chromatography (GPC) (Viscotek Rimax apparatus) in chloroform using calibration on polystyrene standards, at 25 °C with a flow of 1 mL/min, using two ViscotekPolymers columns (Malvern Panalytical Ltd., Malvern, UK) with a refractive detector.

#### 2.7.5. Determination of the Amount of Active Hydroxyl Groups in Obtained Polymers

The presence of hydroxyl groups in the obtained oligomers was estimated using a comparative analysis of ^1^H NMR spectra before and after the reaction with trichloracetylisocyanate in a deuterated chloroform solution.

#### 2.7.6. Wettability

Wettability tests were performed on polymeric films prepared by dissolving 0.5 g of each polymer in 10 mL dichloromethane (DCM), pouring it into a 9 cm diameter glass Petri, dish, and leaving it to evaporate the solvent for 24 h. Tests were conducted on the drop shape analysis system (DSA 25, Kruss, Germany) with the use of ultra-high-quality water (UHQ-water produced in UHQ PS apparatus, Elga, High Wycombe, United Kingdom) using the sessile drop method. SFE was calculated according to the Owens–Wendt equation using water and diiodomethane (Sigma Aldrich, St. Louis, MI, USA) as polar and non-polar liquids, respectively. In each case, 10 drops (0.5 µL in volume) were seeded on the surface of the samples, and the contact angle was measured automatically.

## 3. Results and Discussion

To obtain polyesters with strong antibacterial properties, the synthesis of aliphatic copolyesters containing secondary amine groups in the chain was planned, expecting a comparable strong effect, as in the case described in the literature on aliphatic polyamines [28]. As previously confirmed, it is important to maintain the appropriate hydrophilic/hydrophobic balance property, preserving the hydrophobic part that destroys the lipid membrane of bacterial cells [40]. This effect was achieved by introducing hydrophilic chain sequences and creating a block structure, where hydrophilic and hydrophobic blocks form the main polymeric chain.

The planned polymers were obtained in a two-step process. In the first stage, polyesteramines were synthesized by the polytransesterification of the tartaric acid ester derivative with aminodiols (Table 1, samples from I1 to I5). The final product was then used in the copolymerization reaction of lactide with glycolide as a macroinitiator. After the deprotection of amino and hydroxyl groups, a copolymer containing hydrophobic PLAGA blocks separated by a hydrophilic polyesteramine block with pendant hydroxyl groups was obtained (Table 2, samples from P1 to P8).

### 3.1. Obtaining Polyesteramines and Polyesters-Macroinitiators of Copolymerization of Lactide with Glycolide

Polyesteramines were manufactured by the polytransesterification reaction of dimethyl 2,3-O-Isopropylidene-l-tartrate DIPT (a derivative of dimethyl ester of tartaric acid with protected hydroxyl groups) with aminodiols containing protected secondary amino groups (tert-butyl bis(2-hydroxyethyl) carbamate BocDEA and {(di-tert-butyl ethane-1,2-diylbis[(2-hydroxyethyl) carbamate]} Boc_2_HEDEA (Figure 3).

The obtained polymers present similar properties to polymers manufactured by polycondensation of a tartaric acid derivative with blocked OH groups and hexanediol [41]. To study the effect of amino or hydroxyl groups on the properties of the polymers, we prepared analogous polymers without side hydroxyl groups in the reaction of succinic acid dimethylester (SADE), and without amino groups in the reaction of DIPT with 1,4 butanediol BD for comparison. Finally, we obtained a diversity of polyesters and polyesteramines differing in the composition and amount of hydroxyl or amino groups. This synthesis was carried out by a two-step bulk transesterification method [42,43] with titanium (IV) butoxide as the reaction catalyst. The length of the polyester chain (average molecular mass) was measured with ^1^H NMR by analysis of the amount of OH-CH_2_ end groups. A 7% molar excess of the diol was used relative to the stoichiometric amount of tartaric acid ester derivative. Such an excess made it feasible to obtain polymers where both ends of the chain were generally terminated with hydroxyl groups, which was also accomplished earlier under analogous synthesis conditions in reactions of a similar nature [42].

During the tests of conducting the reaction with the participation of DIPT and aminodiols with protected amino groups, it turned out that raising the reaction temperature above 160 °C was impossible due to the progressive unblocking of hydroxyl and amino groups observed under these conditions. This phenomenon led to the formation of a cross-linked, insoluble, and infusible product. For the above reasons, polymers with high molecular weights have not been possible to obtain this way. We stopped the reaction when the average polyester chain length was about 20–30 units. The characteristics of the obtained polymers are pictured in Table 1.

After 15 h of the reaction of DIPT with 1,4-butanediol, we produced a polymer with the number average molecular weight M_n_ of about 7600 g/mol (estimated based on the intensity of the end group NMR signals). The ^1^H NMR spectra showed the small intensity of A signals of the CH_2_-OH protons, next to the A’ signals of the CH_2_-O-C(CH_3_)_2_ groups (Appendix A), indicating the start of the reaction of spontaneous unprotection of the hydroxyl groups. Based on these signals, the number of units with deprotected OH groups were estimated to be no more than about 5% of the total. The number of methyl protons of tartaric acid ester −COOCH_3_ (Appendix A, signal B) was also negligible, which proves that the chains of the obtained polyester were mainly terminated with OH groups.

The reaction of DIPT with the diethanolamine derivative BocDEA was carried out in a shorter time, obtaining a similar degree of polymerization. The ^1^H NMR spectrum of the product was more complex than previously presented. The signals of the CH_2_ protons of the diethanolamine-derived units (signal a′, b′, Figure 1A) were broader and strongly split. Similarly to the polymerization described earlier, the intensity of A signals related to the number of CH_2_-OH self-deprotected hydroxyl groups was low, not more than 4% of the total. In the analyzed ^1^H NMR spectrum (Figure 1A), as mentioned earlier, a complex signal was observed at about 3.7 ppm, but it may be partially derived from the CH_2_ protons of the chain units adjacent to the deprotected amino groups. The resulting polymer solution was reacted with an excess of trichloracetylisocyanate to confirm the assignment of signals. Only a substantial shift of the a and b signals was observed on the spectrum. This fact established that a and b signals were only from the CH_2_ protons linked to ending chain −OH groups reacted to form urethane bonds. The location of the other peaks has not practically changed. It follows that the self-deprotection of the secondary amines of the chain essentially did not occur.

In the case of polytransesterification of DIPT carried out with Boc_2_HEDEA, the course was similar to the previously discussed reaction with BocDEA. Within 10 h, a similar average degree of polymerization was obtained (DP about 20). In the ^1^H NMR spectrum of this polymer (Figure 2A), in the range of 3.25–3.5 ppm, there was practically only one more series of a″ spectral peaks associated with the presence of O(CO)NCH_2_CH_2_N(CO)O groups in the chain.

Despite twice the amount of protected amino groups, the self-deprotection of these groups under the reaction conditions (Figure 2A, signal A) was also slight.

We supplemented the series of polymers with polyesteramines obtained in the polytransesterification reaction of SADE succinic acid methyl ester with previously used BocDEA or Boc2HEDEA. The polymerization was somewhat faster than the reactions conducted with DIPT (Table 1). The NMR spectra of both products (Figure 1B and Figure 2B) were substantially different from the spectra obtained earlier by the occurrence of the A signal CH_2_ protons of the chain unit derived from succinic acid (at 2.6 ppm).

The presented results showed that, under the reaction conditions, it was possible to obtain all the polymers of sufficient purity and the assumed chain microstructure, permitting them to be applied as a macroinitiator in the planned copolymerization of lactide with glycolide.

### 3.2. Synthesis of Block Terpolymers by Copolymerization of L-Lactide with Glycolide in the Presence of Obtained Polyesteramines or Polyesters as Macroinitiators

To obtain block amphiphilic terpolymers, we copolymerized L-lactide with glycolide using the earlier received polyesteramines and polyesters (Table 1) as the ROP macroinitiator (Figure 4). We used Zirconium (IV) acetylacetonate as a catalyst, as established for this type of reaction [42,44]. We kept a constant molar ratio of L-lactide to glycolide of 80:20, ensuring subsequent solubility of the product. Finally, a series of terpolymers of different microstructures and the length of the hydrophilic block were obtained (Table 2).

By copolymerizing La with GL in the presence of 20 mol.% of poly(butylene 2,3,-O-isopropylidene-L-tartrate) (Table 1, I1, Figure 4A), we obtained the NLH 37 terpolymer (Table 2 NLH 37, Figure 3I) with a composition that is not much different from the theoretical one. The ^1^H NMR spectrum clearly shows the signals associated with glycolidyl (CH_2_-signal G) and lactidyl units of the chain (CH, CH_3_-n, and m signals, respectively). Based on the NMR spectra, the conversion of L-lactide was estimated to be about 93%. No unreacted glycolide was established. The chain’s proton signals assigned to the initiator units were present at the same location as in the starting polyester. However, the attendance of the methylene protons’ A signals indicating unblocking of hydroxyl groups was not observed (Figure 3I). The −CH_2_OH end group signals (Figure 2, signal b) previously seen in this initiator were also absolutely missing. After purification, the obtained terpolymer was dissolved in THF; then, the resulting solution was acidified for hydrolyzing the isopropylidene groups, and the product was precipitated in cold methanol and dried. The spectrum of the terpolymer after deprotection is shown in Figure 3II. The appearance of signal A coming from the protons of the CH_2_OH, as a result of the deprotection with the simultaneous disappearance of signal c and the signal A’ associated with the CH_3_COCH protons of the blocking group, was detected. The carried-out reaction resulted in the deprotection of OH groups reaching 85% of their previous total amount. The wide band appearing in the range of 2.8–3.0 ppm can be attributed to the presence of just-formed hydroxyl groups −OH. To confirm the presence and chemical activity of the unprotected OH pendant groups, the terpolymer solution was additionally reacted with trichloroacetyl isocyanate and then subjected to NMR measurements. The obtained spectrum is shown in Figure 3III.

As expected, as a consequence of the reaction of the hydroxyl groups with the isocyanate, the A signals of the methine protons -(CO)OCHOH were strongly shifted due to the formation of the adjacent urethane group. A similar shift of the R-OH signal at 4.4 ppm was also confirmed, ensuring its good assignment to the protons of the CH_2_ groups neighboring the hydroxyl end groups of the chain. For the same reason, the broad peak of hydroxyl protons previously present in the spectrum of the terpolymer also disappeared. Similar results were obtained during the synthesis of the P2 terpolymer (Table 2. NHL38, Figure 3IV,V). The resulting terpolymer contained a longer block assembled of units derived from tartaric acid (29 mol%). In this terpolymer, more than 90% of hydroxyl groups were deprotected; unfortunately, it was related to the stronger hydrolysis of its ester bonds, causing a large increase in the intensity of the R-OH signal visible on the NMR spectrum (Figure 3V). This phenomenon also caused a decrease in the average molecular weight, which is reflected in the observed much lower inherent viscosity of the terpolymer solution after the deprotection process.

Another kind of terpolymer (Table 2, P3, P4, P5) was obtained during the copolymerization of L-lactide and glycolide with the participation of I2, I3 macroinitiators (Table 1, Figure 4) containing protected hydroxyl and amine groups. In the presence of these initiators, the copolymerization process was significantly slower. A slightly lower efficiency of the copolymerization mainly due to lower lactide conversion affected the final terpolymer compositions, which differed from the theoretical ones. In the ^1^H NMR spectra of terpolymers with deprotected OH and NH groups (Figure 4(AI,BI,BII)), in addition to the signals associated with the CH_2_ protons of the glycolidyl units (signal G), CH and CH_3_ lactidyl units (signals n and m), and the previously discussed −CH_2_ methylene proton signals, derived from the macroinitiator I2 (Figure 4(AI), signals b’ and a’) or I3 (Figure 4(BI), II, signals b’, a’ and a″).

The peaks of methine groups −CHOH adjacent to unblocked hydroxyls were also observed (Figure 4(AI,BI,BII), signal A), as well as signals of the methine protons of the lactidyl units terminating the polymer chain −(CO)(CH_3_)CHOH (Figure 4(AI) signal R-OH). In addition, there were broad signals in the range of 3.0–3.8 ppm associated with the protons of the deprotected −OH and −NH- groups (Figure 4(AI,BI,BII)). The changes in the NMR spectrum observed after the reaction of the solution of these terpolymers with trichloroacetylisocyanate prove the correct assignment of the signals, as in previously discussed studies. The result of the reaction of the protons of the hydroxyl and amino groups with the isocyanate group, the signals of OH, NH, R-OH, and A were shifted outside the observed region (Figure 4(AII)). This unquestionably proves that the terpolymers containing chemically active hydroxyl groups and secondary amino groups in the chain were obtained. The total degree of deprotection of these groups, assessed based on the observation of changes in the intensity of c + g signals, ranged from 75 to 85%.

The last group of terpolymers (Table 2, P6, P7, and P8) was obtained during the copolymerization of lactide with glycolide in the presence of I4 or I5 macroinitiators (Table 1), i.e., containing blocked amino groups.

The copolymerization proceeded with high efficiency—over 90%. The degree of deprotection of amino groups was about 75–82%, which was lower than in the case of deprotection of hydroxyl groups (P1, P2 terpolymers). The obtained spectra of the terpolymers are shown in Figure 5. In the spectra of terpolymers with blocked amino groups, there were signals associated with both lactidyl units (Figure 5(AI) signals n and m) and G-glycolidyl units, as well as those coming from the macroinitiator (signals b’, a’ and a’’ and A, the same as in the starting macroinitiator). However, we did not see b signals from the end groups of the CH_2_-OH initiator (see Figure 2B). After the deprotection of the amino groups, changes in the range of 4.0–2.5 ppm can be seen consisting in the rounding of a’, a’’ and A signals, caused by the superimposition of the broad signal of NH protons on these signals (Figure 5(AII,BI,BII)). After reacting terpolymers with deprotected NH groups with trichloroacetylisocyanate, the shape of these signals returns to a similar state as the initial one, although their slight shift is noticeable (Figure 5(AIII)), likely caused by the influence of the proximity of urea derivatives formed in the reaction of isocyanate with amines.

### 3.3. FTIR Spectroscopy Results

The efficiency of the release of functional groups in terpolymers was also monitored by applying FTIR measurements, and analyzing the changes before and after the deprotection of OH and/or NH groups (Figure 6). The oscillatory spectrum of the terpolymer obtained with the participation of poly(butylene 2,3,-O-isopropylidene-L-tartrate) before and after deprotection of hydroxyl groups is shown in Figure 6(Aa,Ab). The spectrum shows a characteristic strong band of stretching vibrations C=O bonds occurring at 1720 cm^−1^, derived from ester groups in the terpolymer; after the process of deprotection, it widens. The presence of vibration signals in the 3600–3400 cm^−1^ allocation, attributed to O-H stretching vibrations Figure 6(Ab), proves the effectiveness of deprotection and the presence of hydroxyl groups in the final terpolymer. Additionally, the removal of the isopropyl group blocking the hydroxyl groups is confirmed by the decrease in the intensity of the δCH_3_ scissor vibrations occurring in the area of 1450 cm^−1^. This phenomenon is also evidenced by the changes observed in the range of 1200–1000 cm^−1^.

In turn, the spectrum showing changes during the deblocking of amino groups in the spectrum of the terpolymer obtained in poly{(di-tert-butyl ethane-1,2-diylbis[(2-hydroxyethyl)carbamate] succinate)} was shown in Figure 6(Ba,Bb). After the deprotection process, the appearance of a strong absorption band in the range of 3200–3500 cm^−1^ was observed (Figure 6(Bb)). This effect is related to the vibrations stretching N-H bonds of secondary amine groups. Additionally, a newly noted band at 1640 cm^−1^ can be attributed to vibrations deforming the bonds of secondary amine groups, and a band at 1190 cm^−1^ is typical of C-N bonds of aliphatic secondary amines. The success of the deprotection reaction of amino groups is also evidenced by the disappearance of the band in the range of 1650 cm^−1^ assigned to the carbamide blocking group with the simultaneous decrease in intensity of the band −C(CH_3_)_3_ groups.

The most difficult to interpret were changes in the spectra of terpolymers containing both types of functional groups (Figure 6C Analysis of the FTIR spectra in these terpolymers confirmed the existence of O-H and N-H stretching vibration signals ranging from 3500–3100 cm^−1^. Additional confirmations of the removal of protecting groups are the phenomena described earlier, i.e., a decrease in signal intensity at 1450 cm^−1^ and 1385 cm^−1^, the disappearance of the band at 1660 cm^−1^ coming from carbamide groups, and the formation of a characteristic band at 1640 cm^−1^ belonging to secondary amino groups. However, these changes are much less observable and obvious compared to the changes taking place in the previously discussed terpolymers. In the case of the presence of amino and hydroxyl groups, the noticeable smearing and band broadening following the deprotection of functional groups may be due to the overlap of vibrations of partially unblocked and deprotected groups.

### 3.4. Static Contact Angle Measurements

A spectacular effect of the presence of hydroxyl and secondary amine groups in the obtained triblock copolymers involved changes in their behavior in the aqueous environment. Films made of terpolymers were assessed by determining the static contact angle of their surface by a drop of water (sample image—Figure 6D). The results are presented in Table 2. The majority of obtained terpolymers with blocked functional groups showed the value of the static contact angle to be remarkably similar to the L- lactide/glycolide (85/15) copolymer, which is about 80°–82°. Only the surface of terpolymers containing both blocked hydroxyl and amino groups (P3, P4, and P5) was better wettable by water, which was related to the presence of protected groups in the form of carbamate and dioxolane derivatives. The terpolymer possessing most of such groups had the smallest contact angle of 62° After the deprotection of the functional groups, all tested terpolymers had significantly higher wettability. For terpolymers containing only side hydroxyl groups (P1, P2), the tested contact angle increased with the length of the hydrophilic block and was 53° and 50°, respectively. In the case of terpolymers containing secondary amino groups in the chain (P6, P7, and P8), an increase in amine content in the chain affected an even stronger rise in wettability (contact angles were respectively; 49°, 36° and 34°). It is not surprising then that the terpolymer P5 containing the greatest amount of amine and hydroxyl groups showed the smallest contact angle, of less than 20°.

### 3.5. Average Molecular Weight Measurements

To estimate the average molecular weight of the obtained terpolymers and the profile of their mass dispersion, their solutions were subjected to GPC tests, and their inherent viscosities were determined. All the obtained GPC elugrams showed a monodisperse molecular weight distribution, whether they were materials before or after the deprotection process. These results confirmed the earlier conclusions that the products obtained were indeed block terpolymers and not a blend of polymers. Examples of GPC elugrams are shown in Appendix A. The weight-average molecular weights of the terpolymers not subjected to the functional group deprotection reaction were similar and ranged from about 16,000 g/mol to nearly 21,000 g/mol (Table 2). These values were lower than theoretically calculated from the M/I molar ratio. The determined values of inherent viscosity oscillated from 0.25 dL/g to 0.46 dL/g. The mass distributions ranged from 2.6 to 4.5. They were excessive, but within the expected limits, taking into consideration that the used macroinitiators were obtained by step polymerization. As for the results of GPC tests of products obtained after functional groups deprotection, the obtained values of average molecular mass M_w_ were lower and oscillated from about 5000 g/mol to less than 15,000 g/mol. The main reason for such low values was likely the effect of the interaction of amine and hydroxyl functional groups with the packing of the columns, resulting in the extension of retention time, and calibration with the use of polystyrene standards probably distorted the final results. However, there is undoubtedly a decrease in the size of the molecular weights. It was a consequence of the partial hydrolysis of the terpolymer ester bonds during the deprotection reaction, causing the observed increase in mass dispersion even to the value of Đ above 4.5 in the case of the P4 terpolymer.

### 3.6. Thermal Properties of the Terpolymers

To determine the semi-crystallinity of the obtained materials, and their glass transition temperature, its samples were subjected to DSC tests. In Table 2 and Figure 7, the results are presented. The glass transition temperatures of the obtained terpolymers ranged from 22 °C to 45 °C, depending on the composition. Thus, they were lower than in the case of the analogous copolymer of L-lactide with glycolide (85:15) obtained with Zr(acac)_4_ [45], where the temperature was 58 °C. The observed difference was affected by the much lower molecular weight of the tested terpolymers, and also by the presence of polyester blocks containing functional groups. In none of the tested terpolymers, even in materials containing a polyester block derived from the initiator of more than 20 mol.%, the occurrence of the second glass transition temperature was not observed.

The deprotection of functional groups did not significantly impact the change of the glass transition temperature in most cases. The DSC thermogram of L-lactide/glycolide copolymer pictured only one large melting endotherm of the crystalline regions of the lactidyl block (about 41 J/g, melting point above 140 °C) [45]. In the tested terpolymers, the lactidyl block was able to form larger crystal areas practically only when the protected groups were removed. The endotherm responsible for the melting of these semicrystalline regions appeared in the range from about 90 °C to even close to 170 °C, with a heat of fusion from about 15 J/mol to over 60 J/mol. The second, much smaller endotherm, appearing only for a few terpolymers in the range of 40–80 °C, most likely originated from the crystalline areas of the terpolymer chain block derived from the macroinitiator. A relatively high degree of crystallinity was exhibited by terpolymers containing unblocked amino groups, capable of forming strong intra- and intermolecular bonds, similar to polymer blends [46]. Similar crystalline areas associated with the semicrystal structure formed by a regular hydrogen bonding network (T_m_ about 160 °C, melting enthalpy over 34 J/g) were observed in aliphatic polyesteramides obtained by polycondensation of α,ω-aminoalcohols with dimethyl suberate and dimethyl sebacate [47].

## 4. Conclusions

As described, the method of copolymerization of L-lactide with glycolide, initiated with selected macroinitiators, makes it possible to obtain a series of biodegradable block amphiphilic terpolymers containing in the chain hydrophobic blocks composed of lactidyl and glycolidyl units and a hydrophilic block possessing side hydroxyl and/or amino groups. Thus, it is possible to control the properties of the final copolymer by selecting both the composition of the macroinitiator and its amount. However, this method has limitations, mainly related to the first stage of the macroinitiator synthesis. The polytransesterification process must be carried out in very “delicate” conditions, at a relatively low maximum temperature of 160 °C and for a limited time. Therefore, it is practically impossible to obtain polyesteramines with high molecular weight by this method. The purity of the polyesteramines was sufficient to be used as ROP initiators of L-lactide and glycolide copolymerization in the presence of a coordination catalyst, i.e., zirconium (IV) acetylacetonate. Terpolymers were obtained with high yield and composition close to the assumed one. NMR studies showed that the terpolymer chain block formed by the units of macroinitiator derivative was not terminated with OH groups, and did not form chain ends. The distribution of molecular weights was monodispersed, and only one glass transition temperature was present in the thermographs. All this proves that indeed a block terpolymer was obtained, and not a blend of two polymers.

A serious problem encountered during the preparation of these functional terpolymers was the process of deprotection of hydroxyl and amino groups, especially when they were present simultaneously in the final terpolymer chain (P3, 52, 56). The applied universal method of acidic hydrolysis of protecting groups allowed the removal of the carbamate and dioxolane protecting groups during one procedure but resulted moreover in the hydrolysis of some ester bonds. This phenomenon caused a significant decrease in the average molecular weight and an increase in the mass dispersion of the final terpolymers. The effect of this process was a substantial increase in the wettability of the surface of the films made of synthesized terpolymers containing unprotected hydroxyl and/or amino groups. The decrease in the value of the water contact angle is particularly strong in terpolymers containing amino groups. Depending on the length of the hydrophilic block and its composition, the angle value ranged from 50° to 34°.

Polyesters with active functional groups allow for fairly easy modifications with various biomolecules, which can lead to the production of biomaterials with the possibility of controlled interactions with cells. Maintaining a balance between hydrophobicity and hydrophilicity is essential to ensure protein adsorption and cell growth. The obtained terpolymers exhibit such a property (contact angle of about 50°), and additionally, the degree of wettability of their surface can be controlled to a large extent by the selecting of composition. Summing up, the obtained materials seem to be particularly interesting as a material for use in tissue engineering—of course provided that other essential properties such as biocompatibility, or the appropriate degradation profile, will be fully adequate. The results of antibacterial activity, cytocompatibility, and biodegradation tests will be a matter of a separate publication. It also seems interesting to use the amphiphilic properties of the described materials in the creation of micro- and nanocarriers for drugs. These polymers were also synthesized to use them as carriers of biologically active compounds for cosmetics or dermatology. They will be involved in the gradual release of these compounds into the skin while supplying protection against bacterial contamination.

## Data Availability

The data presented in this study are available on request from the corresponding author.

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
