# Peer review of "Biodegradable Block Poly(ester amine)s with Pendant Hydroxyl Groups for Biomedical Applications"

_polymers, 2023, doi:10.3390/polym15061473_

Round 1
Reviewer 1 Report (Previous Reviewer 1)
The authors have addressed the reviewer’s comments, so it can be accepted in the present form.
Author Response
Reviewer 1.
The authors have addressed the reviewer’s comments, so it can be accepted in the present form.
Thank you for your evaluation and help in improving our manuscript.
Reviewer 2 Report (Previous Reviewer 2)
Overall, this manuscript demonstrates the amphiphilic block terpolymers. The synthetic conditions were explored and optimized. The obtained polymers were well characterized by different techniques. Compared to the previous version, the manuscript has been greatly improved. The This work is worthy of publication in Polymers. However, the following minor concerns should be addressed before publication.
1. Please check all the abbreviations in the main text. E.g., Page 4, Line 186, should be (BOC)2O. Please keep identical format, and it should be moved to Line 179 since it was the first time to be mentioned.
2. Tables 1 and 2, what is the naming system of the polymers? There is no explanation in the main text, please indicate. It is a little bit confusing that the numbers are not in order.
3. Figure 6D, the water contact angle experiment, please add labels such as arrows and actual values (degree).
4. The paper indicated that the synthesized polymers could be applied for biomedical applications. It is good to see the real data for those applications in the future work.
Author Response
Reviewer 2.
Overall, this manuscript demonstrates the amphiphilic block terpolymers. The synthetic conditions were explored and optimized. The obtained polymers were well characterized by different techniques. Compared to the previous version, the manuscript has been greatly improved. The This work is worthy of publication in Polymers. However, the following minor concerns should be addressed before publication.
Thank you for your review and comments, which certainly improved the quality of our manuscript. We have made changes and corrections following all your comments.
- Please check all the abbreviations in the main text. E.g., Page 4, Line 186, should be (BOC)2 Please keep the identical format, and it should be moved to Line 179 since it was the first time to be mentioned.
We did not notice this error, we corrected everything according to your comment. In all the abbreviations of di-tert-butyl di-carbonate we entered the correct version - (Boc)2O. We also checked the correctness of other abbreviations we used.
- Tables 1 and 2, what is the naming system of the polymers? There is no explanation in the main text, please indicate. It is a little bit confusing that the numbers are not in order.
Of course, you are right, we changed the numbers for both the initiator polymer samples (Table 1, I1, I2, I3, etc.) and the obtained terpolymers (Table 2, P1,P2,P3, etc.). We also introduced changes in the manuscript's content, replacing the names of these polymers per the introduced nomenclature.
- Figure 6D, the water contact angle experiment, please add labels such as arrows and actual values (degree).
We introduced these elements into the mentioned Figure 6D, marked the angles between the tangent to the drop and the ground, and entered the values of these angles.
- The paper indicated that the synthesized polymers could be applied for biomedical applications. It is good to see the real data for those applications in the future work.
As we write in the Conclusions section, the synthesis of the described compounds was intentional from the beginning, the goal is to obtain new carrier materials for active substances, at the same time protecting against bacterial contamination, which will be a component of cosmetic creams and dermatological ointments. We are also trying to use selected terpolymers as a material for forming scaffolds for cell culture. We already have promising results from bactericidal and cell toxicity studies, which we want to publish soon.
This manuscript is a resubmission of an earlier submission. The following is a list of the peer review reports and author responses from that submission.
Round 1
Reviewer 1 Report
The manuscript "Biodegradable block poly(ester amines) with pendant hydroxyl 2 groups for biomedical applications" presents a simple and reliable method to obtain a new group of biodegradable block co-polyesters containing side hydroxyl and secondary amino groups in the chain. It presents issues related to the synthesis and research of physicochemical properties of the obtained block. The idea seems to be novel and the experiments were carried out nicely. However, the manuscript has shortcomings that prevent its publication in the current form:
-The abstract part: I think it would be better to add the most important findings to this part (add the significant results).
- Please revise the numbers in Table 2.
- The quality of figures 10 and 12 needs to improved.
-I suggest these references to be included in this manuscript.
10.3390/polym14173697
10.3390/polym14010215
Reviewer 2 Report
Overall, this manuscript is not a decent one. The data are poorly presented. Unfortunately, this work is not worthy of publication in Polymers currently since the journal is requiring high quality of draft. There are some issues to be addressed (see the following concerns). However, if the authors can revise the manuscript and greatly improve the data presentation, it can be considered when it is re-submitted.
1. The figures in the manuscript are poorly prepared, although the data looks fine. All the chemical structures should be drawn properly in Chemdraw using the same template. The labels in the figures should use identical size and font. The panel captions (e.g., A, B, C…) should be on top-left corner of the panels and well aligned.
2. There are too many figures in the main manuscript which is not appropriate. Please consider moving some data (e.g., NMR spectra) into supplementary materials.
2. Figure legend in Fig. 11 seems wrong. They are even not written in English! Some parts of the figures are cropped off.
3. Scheme 2 and 3, the reaction conditions are missing. Please carefully check all the description on the reactions.
4. In Figure 10, what are a) and b) in each FTIR spectrum? They are not indicated.
5. Line 35-39, the authors introduced various types of bioresorbable temporary implants. Please add references accordingly.